# Inhibiting Myostatin Expression by the Antisense Oligonucleotides Improves Muscle Wasting in a Chronic Kidney Disease Mouse Model

**DOI:** 10.3390/ijms26073098

**Published:** 2025-03-27

**Authors:** Arju Akhter, Abdullah Md. Sheikh, Jun Yoshino, Takeshi Kanda, Atsushi Nagai, Masafumi Matsuo, Shozo Yano

**Affiliations:** 1Department of Laboratory Medicine, Faculty of Medicine, Shimane University, 89-1 Enya-Cho, Izumo 693-8501, Japan; arjuakhter13@gmail.com (A.A.); abdullah@med.shimane-u.ac.jp (A.M.S.); 2Department of Nephrology, Faculty of Medicine, Shimane University, 89-1 Enya-Cho, Izumo 693-8501, Japan; 3The Center for Integrated Kidney Research and Advance (IKRA), Faculty of Medicine, Shimane University, 89-1 Enya-Cho, Izumo 693-8501, Japan; 4Department of Neurology, Faculty of Medicine, Shimane University, 89-1 Enya-Cho, Izumo 693-8501, Japan; anagai@med.shimane-u.ac.jp; 5Graduate School of Science, Technology and Innovation, Kobe University, Kobe 657-8501, Japan

**Keywords:** sarcopenia, skeletal muscle, chronic kidney disease, myostatin, antisense oligonucleotide

## Abstract

Sarcopenia, a serious consequence of chronic kidney disease (CKD), is driven by elevated myostatin (MSTN), a key inhibitor of muscle growth. This study explored the potential of an MSTN-specific antisense oligonucleotide (ASO) in reversing CKD-induced muscle wasting in a mouse model. Thirty-two male C57BL/6J mice were randomly assigned to a non-CKD group (n = 8, regular diet) and a CKD group (n = 24, adenine diet). CKD was induced using a 0.2% adenine-supplemented diet for 4 weeks. Following this, the mice were sub-grouped into CKD (saline, n = 8), CKD + Low-Dose ASO (25 mg/kg ASO, n = 8), and CKD + High-Dose ASO (50 mg/kg ASO, n = 8). ASO was administered via subcutaneous injections for 8 weeks. Muscle mass, treadmill performance, grip strength, and muscle fiber morphology were assessed alongside qPCR and Western blot analysis for MSTN, atrogin-1, and MuRF-1 expression. ASO therapy significantly enhanced muscle mass and function and enlarged muscle fibers while effectively downregulating muscle degradation markers. These improvements occurred without compromising renal function, as confirmed by BUN, creatinine, kidney weight, and histological analysis. This study is the first to demonstrate the efficacy of ASO therapy in mitigating CKD-induced sarcopenia, offering a promising targeted gene therapy with significant clinical implications for improving nutritional status and physical performance in CKD.

## 1. Introduction

Sarcopenia, a degenerative phenomenon characterized by a sequential erosion of muscle composition and functional capacity, frequently raises a significant concern within the chronic kidney disease (CKD) population [1,2]. However, the rate of its occurrence varies depending on CKD stages and sociodemographic and health status [3,4]. The adverse outcomes of progressive muscle atrophy in CKD patients are significant as it accelerates the lowering of muscle strength and physical performance and raises morbidity and mortality [5,6]. Muscle atrophy not only lowers our quality of life but also increases our dependency on healthcare systems, provoking new global health and economic challenges [7,8]. While a few management regimens—such as nutritional support, including calories, protein, vitamin D supplementation, minerals, and dietary fibers; correction of metabolic acidosis with sodium bicarbonate; physical exercise; and modulation of gut microbiota through probiotics, prebiotics, and symbiotics—provide some benefits to address sarcopenia in CKD individuals, emerging pharmacological approaches like anabolic agents, myostatin (MSTN) inhibitors, and anti-inflammatory treatments are still under investigation [9,10,11,12]. These current therapies for muscle atrophy in CKD are not sufficient to halt the progression of sarcopenia due to its complex pathophysiology [13]. At present, this urgent clinical problem raises a significant concern and drives the demand for new therapeutic targets. Therefore, the development of new therapies for CKD-associated muscle atrophy with improved efficacy and safety is urgently needed.

Several pathological conditions, such as chronic inflammation, oxidative stress, increased uremic toxins, negative protein balance (increased protein degradation and reduced protein synthesis), insulin resistance, and autophagy, are the major contributing factors to sarcopenia in CKD [14,15]. MSTN signaling pathway activation is a potential mechanism suggested for underlying sarcopenia in CKD [16]. MSTN, scientifically recognized as Growth Differentiation Factor-8 (GDF-8), restricts the potential of skeletal muscle growth, which induces protein breakdown and skeletal muscle loss by activating the ubiquitin–proteasome pathway [17,18]. Moreover, ubiquitin ligases primarily associated with muscle tissue, for example, Atrogin-1, also termed F-box gene 1, in conjunction with MuRF-1, alternatively called Muscle RING-finger protein-1, are involved in promoting proteasomal degradation of muscle proteins, especially myofibrillar proteins known as myosin heavy chains, during the progression of CKD [19]. On the contrary, *MSTN* gene knockout, mutation, or its inactivation contributes to muscular hypertrophy in humans and animals, such as mice, cattle, sheep, dogs, cows, and pigs, without promoting any disease or having any harmful effect on the quality of life [20,21,22,23,24,25,26,27,28].

Several studies investigated MSTN inhibition’s ability to promote muscular strength and functionality in CKD, exploring its benefits as a therapeutic target for preventing sarcopenia [29,30,31]. Recently, antisense oligonucleotides (ASOs) have attracted considerable interest in the area of drug development and have shown promise to alter the expression of disease-related genes, including the MSTN gene. For instance, Eilers et al. demonstrated that targeting MSTN mRNA with phosphorodiamidate morpholino oligomers (PMOs) decreased MSTN levels and subsequent muscle growth in animal models [32]. Similarly, Zhou et al. reported that myostatin inhibition synergistically enhanced the effects of Survival motor neuron 1-restoring antisense therapy, leading to increased muscle mass, improved motor function, and extended survival in spinal muscular atrophy (SMA) mice [33]. ASO-mediated exon-skipping treatment is also clinically implicated in preventing Duchenne Muscular Dystrophy (DMD) [34]. Moreover, the exon-skipping mechanism of ASO was studied for its potential to suppress MSTN gene function [35]. Despite these promising findings, in the context of CKD, few ASO-based antisense anti-CKD drugs have been developed, and the impact of ASO therapy on muscle atrophy in CKD is not thoroughly understood [36].

In this study, we sought to investigate the therapeutic potential of KMM001, an 18-mer chimeric ASO against the MSTN gene (in the form of a novel myostatin-specific inhibitor synthesized by KNC Laboratories Co, Ltd. (Kobe, Japan)), in boosting muscular development, strength, and efficacy within an animal model of CKD. Previous in vitro studies illustrated that inhibiting the MSTN gene via KMM001 effectively decreased mature MSTN mRNA and protein levels in CRL-2061 cells. This ASO was also reported to exhibit enhanced human myoblast proliferation [37]. However, the in vivo effects of KMM001 in treating muscle-wasting conditions using a mouse model remain unexplored. In this study, we designed and implemented an experimental mouse model mimicking muscle atrophy driven by CKD and then investigated the effects of MSTN-ASO (KMM001) on CKD-induced muscle atrophy. Our research presents new insight into the molecular mechanism and the potential of gene-targeted therapies as a variable intervention for muscle preservation in CKD.

## 2. Results

### 2.1. MSTN-ASO Treatment in Mice Increases Skeletal Muscle Mass, Strength, and Function

To investigate the therapeutic potential of MSTN-ASO in counteracting CKD-induced muscle atrophy, we administered MSTN-ASO subcutaneously to C57BL/6j mice. MSTN-ASO was dissolved in 0.9% physiological saline and delivered weekly at doses of 25 mg/kg (CKD + Low-Dose ASO group) and 50 mg/kg (CKD + High-Dose ASO group) for 8 weeks (total eight doses, one dose per week). A CKD group receiving saline only served as a disease control, while a non-CKD group receiving a regular diet acted as a healthy control. At the end of the experiment, we assessed the weights of key hindlimb muscles such as gastrocnemius (GC), tibialis anterior (TA), and soleus muscles. Although we did not evaluate liver and cardiac function, no findings such as jaundice or hepatomegaly and no significant difference in the cardiac size among the non-CKD, CKD, and ASO treatment groups were observed. Additionally, no significant increase in body weight was observed in the treatment groups compared to CKD groups (see Appendix A). As illustrated in Figure 1A, the CKD group experienced a remarkable decline in individual muscle weights relative to the non-CKD group (GC: CKD: 0.115 ± 0.014 g vs. non-CKD: 0.161 ± 0.004 g, *p* < 0.001, TA: CKD: 0.033 ± 0.003 g vs. non-CKD: 0.054 ± 0.003 g, *p* < 0.001, Soleus: CKD: 008 ± 001 g vs. non-CKD: 0.013 ± 0.002 g, *p* < 0.001), which is consistent with previous findings that a marked decrease in muscle mass was observed in mice following several weeks on a diet supplemented with 0.2% adenine [38,39]. Similarly, a widely used surgical CKD model, 5/6 nephrectomy, has been shown to induce skeletal muscle loss [40]. Both models consistently demonstrated that CKD progression is strongly related to the development of muscle loss due to the combined effect of uremia, oxidative stress, and systemic inflammation. However, both low doses and high doses of MSTN-ASO exhibited significant increases in all muscle weights relative to the untreated CKD group (GC: CKD + Low-Dose ASO: 0.146 ± 0.006 g, CKD + High-Dose ASO: 0.146 ± 0.007 g, vs. CKD: 0.115 ± 0.014 g, *p* < 0.001, TA: CKD + Low-Dose ASO: 0.050 ± 0.034 g, CKD + High-Dose ASO: 0.050 ± 0.034 g, vs. CKD: 0.033 ± 0.003 g, *p* < 0.001, Soleus: CKD + Low-Dose ASO: 0.010 ± 0.001 g, CKD + High-Dose ASO: 0.011 ± 0.002 g, vs. CKD: 008 ± 001 g, *p* < 0.01), suggesting that CKD-provoked muscle loss was effectively restored by MSTN-ASO administration even at low doses.

Then, we assessed functional outcomes to evaluate whether MSTN-ASO treatment enhanced muscle performance. Although CKD mice developed compromised muscle function after 4 weeks of adenine treatment as reflected by low treadmill endurance and grip strength (see Appendix A), there was no significant improvement in endurance or physical capacity after 3 weeks of MSTN-ASO treatment (see Appendix A). However, after 8 weeks, CKD mice with both low and high doses of MSTN-ASO treatment exhibited significantly improved endurance and physical capacity compared to untreated CKD mice. As shown in Figure 1B, the exhaust time (left), speed (middle), and distance (right) of MSTN-ASO-treated CKD mice were significantly longer or faster than those of CKD mice (Exhaust time: CKD + Low-Dose ASO: 44 ± 4.4 min, CKD + High-Dose ASO: 44 ± 3.1 min, vs. CKD: 35 ± 8.8 min, *p* < 0.05, Speed: CKD + Low-Dose ASO: 171 ± 16.4 rpm, CKD + High-Dose ASO: 173 ± 8.9 rpm, vs. CKD: 141 ± 29 rpm, *p* < 0.05, Distance: CKD + Low-Dose ASO: 1189 ± 229 m, CKD + High-Dose ASO: 1185 ± 140 m, vs. CKD: 812 ± 367 m, *p* < 0.05), indicating a robust improvement in overall stamina.

To further assess muscle strength, we performed a peak forelimb grip test (Figure 1C). Although the 3-week administration of both doses of MSTN-ASO did not significantly enhance forelimb grip strength (see Appendix A), the 8-week treatment significantly enhanced forelimb grip strength in the MSNT-ASO-treated mice compared to the CKD mice (CKD + Low-Dose ASO: 0.14 ± 007 kg, CKD + High-Dose ASO: 0.14 ± 0.003 kg, vs. CKD: 0.08 ± 0.007 kg), *p* < 0.001). These results corroborate previous findings demonstrating that an MSTN blockade enhances not only muscle mass but also functional strength [41,42]. Our findings thus suggest that MSTN-ASO therapy provides both morphological and functional benefits in the context of CKD-induced muscle wasting.

### 2.2. Mstn-Aso Treatment Improves Muscle Atrophy Induced by CKD

To evaluate the inhibitory effects of MSTN-ASO on CKD-induced muscle atrophy, we measured the mean cross-sectional area (CSA) of the GC muscle fibers. As depicted in Figure 2A,B, and Appendix A, CKD mice showed a significant reduction in the average CSA of GC myofibers compared to non-CKD controls (*p* < 0.001). This reduction in muscle fiber CSA was consistent with previous studies indicating the catabolic effects of CKD on skeletal muscle, leading to muscle atrophy [43]. MSTN-ASO treatment significantly increased the CSA in both low (25 mg/kg) and high (50 mg/kg)-dose groups (*p* < 0.01), demonstrating the efficacy of MSTN-ASO in reversing muscle atrophy. Histological analysis also revealed a normal muscle fiber architecture in MSTN-ASO-treated mice, like non-CKD controls. These findings resembled previous research, demonstrating that myostatin inhibition significantly increases muscle fiber CSA in various models of muscle atrophy [44,45].

In addition to the average CSA, we analyzed the distribution of muscle fiber sizes. As shown in Figure 2C, the CKD group exhibited a distinct leftward shift in the muscle fiber size distribution compared to non-CKD controls, indicating a predominance of smaller, atrophic fibers. This leftward shift is characteristic of CKD-induced muscle wasting and aligns with previous reports showing muscle fiber shrinkage in CKD [46]. MSTN-ASO treatment reversed this effect, inducing a rightward shift in muscle fiber size distribution in both low- and high-dose groups, reflecting a shift toward larger myofibers. Notably, the distribution pattern of fiber size in the low-dose and high-dose groups closely resembled that of the non-CKD group. This result suggests that MSTN-ASO treatment is not only effective in preventing muscle wasting but also in accelerating hypertrophy, consistent with the observation from prior studies on myostatin blockade [47,48]. These results confirm that MSTN-ASO effectively restores muscle fiber size and mitigates CKD-induced muscle atrophy.

### 2.3. Atrophic Gene mRNA and Protein Expression Are Suppressed by MSTN-ASO Injections in Mice Skeletal Muscle

To confirm the effective inhibition of myostatin by MSTN-ASO administration, we analyzed the transcriptional levels of essential genes involved in atrophy, including myostatin, Atrogin-1, and MuRF-1, which are critical regulators of protein degradation in skeletal muscle. Among CKD-modeled mice, myostatin, Atrogin-1, and MuRF-1 demonstrated a pronounced upregulation in the gastrocnemius muscle, compared to those of non-CKD mice (*p* < 0.001) (Figure 3A). Aligning with previous studies, these findings demonstrated that CKD activates proteolytic pathways, mainly dominated by the ubiquitin–proteasome system, fostering E3 ligases, such as Atrogin-1, coupled with MuRF-1 elevation and catabolic breakdown of muscle protein [49].

The subcutaneous administration of low and high (25 mg/kg and 50 mg/kg) doses of MSTN-ASO over an 8-week period resulted in a significant downregulation of these genes at mRNA levels compared to untreated CKD mice (*p* < 0.001; *p* < 0.01). This suppression suggests that MSTN-ASO effectively inhibits myostatin signaling and downstream proteolytic pathways, thereby reducing muscle protein breakdown. Similar findings were highlighted by previous studies, where inhibition of myostatin led to decreased expression of Atrogin-1 and MuRF-1 and subsequent muscle mass preservation [50].

To further validate these results at the protein level, we performed a Western blot to quantify myostatin, Atrogin-1, and MuRF-1 protein levels. In addition, we examined the Collagen-1 (Col-I) protein level, which might represent another mature isoform or fragment of Col-I, known as COL1A1, as a fibrosis marker. Consistent with the mRNA data, in the gastrocnemius (GC) muscle of untreated CKD mice, there were significantly elevated protein levels of myostatin, Atrogin-1, MuRF-1, and Collagen-1 (Figure 3B–D). These increases were suppressed to the levels of non-CKD mice by the MSTN-ASO treatment in both dose groups. This result indicates that MSTN-ASO prevents muscle protein breakdown and reduces fibrosis, a key pathological feature in CKD-induced muscle wasting [51]. In summary, MSTN-ASO effectively alleviates CKD-induced muscle atrophy and fibrosis in the muscle, promoting muscle preservation and highlighting its therapeutic potential for CKD-induced muscle wasting.

### 2.4. MSTN-ASO Treatment Does Not Affect Kidney Functions

To assess the impact of MSTN-ASO on renal health, we measured kidney weight and plasma levels of urea nitrogen (BUN) and creatinine after 8 weeks of treatment. In CKD mice, kidney weight was significantly reduced compared to non-CKD mice (CKD: 0.104 ± 0.14 g vs. non-CKD: 0.155 ± 0.012 g, *p* < 0.001), while BUN and creatinine levels were markedly elevated (BUN: CKD: 78.42 ± 21.48 mg/dL vs. non-CKD: 25.30 ± 9.98 mg/dL, *p* < 0.001, creatinine: CKD: 0.37 ± 0.13 mg/dL vs. non-CKD: 0.17 ± 0.03 mg/dL, *p* < 0.001), indicating impaired renal function (Figure 4A). These results show a similarity with previous studies demonstrating that adenine-induced CKD models exhibit a reduced kidney mass and increased markers of renal dysfunction [52,53].

Notably, kidney weight, BUN, and creatinine levels in the CKD + Low-Dose ASO group and CKD + High-Dose ASO group showed no significant differences from the untreated CKD group following MSTN-ASO treatment at 25 mg/kg and 50 mg/kg (kidney weight: CKD + Low-Dose ASO: 0.098 ± 0.021 g, CKD +High-Dose ASO: 0.094 ± 0.013 g, vs. CKD: 0.104 ± 0.014 g, no significance, BUN: CKD +Low-Dose ASO: 76.08 ± 16.32 mg/dL, CKD + High-Dose ASO: 70.18 ± 15.84, vs. CKD:78.42 ± 21.48 mg/dL, no significance, creatinine: CKD + Low-Dose ASO: 0.36 ± 0.07 mg/dL, CKD + High-Dose ASO: 0.32 ± 0.07 mg/dL, vs. CKD: CKD: 0.37 ± 0.13 mg/dL, no significance), suggesting that MSTN-ASO does not mitigate renal dysfunction caused by adenine administration. In contrast, a study showed that Paeoniflorin treatment significantly reduced BUN and creatinine in CKD-induced muscle atrophy models [54].

We further evaluated the histopathological effect of MSTN-ASO on kidneys, which supported these results. HE- and Azan-stained sections of the non-CKD group showed a normal renal architecture, including intact glomeruli, well-preserved brush borders, and the regular arrangement of renal tubules (Figure 4B). In contrast, kidneys from CKD mice exhibited extensive pathological alterations, such as glomerular atrophy, tubular dilation, and brush border loss, which are hallmark features of adenine-induced nephropathy [52,53] (p. 8). This pathological alteration indicated that 0.2% adenine is sufficient to produce CKD irreversibly, and after the 4 weeks of a 0.2% adenine-supplemented diet, switching to a regular diet does not reverse the CKD condition [55]. However, these histological changes persisted in MSTN-ASO-treated groups, indicating no observable improvement in renal architecture.

In addition, Azan staining revealed an extensive deposition of collagen fibers in glomerular tufts, renal tubules, and interstitial tissues in CKD kidneys (Figure 4B,C). The extent of collagen accumulation was similar in the CKD, CKD + Low-Dose ASO, and CKD + High-Dose ASO groups, with no significant reduction in fibrotic areas following MSTN-ASO treatment. Interestingly, while inhibiting MSTN has been implicated in improving fibrosis in skeletal muscle [56], its inhibition in this context did not reduce renal fibrosis. Additionally, others showed the protective effects of Chinese herbs on both CKD-induced muscle atrophy and renal dysfunction through inhibiting oxidative stress, mitochondrial dysfunction, and ferroptosis [54,57] (p. 8). In contrast, MSTN-ASO treatment effectively addresses CKD-induced muscle atrophy via specific inhibition of the *MSTN* gene.

## 3. Discussion

This research provides strong evidence for the efficacy of subcutaneous injection of MSTN-ASO (KMM001) as a novel, targeted, and *MSTN* gene-specific approach in mitigating CKD-induced muscle-wasting conditions. Using an established adenine-induced CKD model in 8-week-old C57BL/6j mice, we demonstrated that MSTN-ASO significantly restored skeletal muscle mass, strength, and function, key markers of muscle health compromised by CKD. Notably, these findings highlight the potential of MSTN-ASO as a gene-targeted therapy for CKD-associated sarcopenia, a condition for which current treatment options remain limited.

As a recognized model, we used 8-week-old C57BL/6j mice to induce muscle wasting in CKD using a 0.2% adenine-supplemented diet, one of the main complications of CKD. On the other hand, to compare the effects under normal physiological conditions, we used healthy control mice grouped as non-CKD without adenine and CKD. After targeting and inhibiting the function of the MSTN gene in the CKD mice followed by subcutaneous administration of MSTN-ASO weekly for eight weeks (after application of a total of eight doses), our study revealed that subcutaneous administration of MSTN-ASO over eight weeks led to a restoration of muscle mass, strength, and function, as confirmed by the weight of skeletal muscles, treadmill test, and forelimb grip strength test, indicating functional recovery of the skeletal muscles. We observed a considerable enhancement of the muscle fiber CSA after histological analysis alongside a rightward shift in fiber size distribution in the treatment group. This indicated the preservation of muscle mass and its functional recovery, a sign of improving muscle wasting in response to MSTN-ASO therapy. Our results resembled previous studies, showing similar benefits of MSTN inhibition in promoting muscle fiber growth and preserving function in various muscle-wasting conditions such as diabetes, cachexia, and obesity [58,59,60,61].

At the molecular level, significant overexpression of myostatin, Atrogin-1, and MuRF was identified in CKD mice, encompassing both mRNA transcripts and protein products, as confirmed by qPCR and Western blotting. This ASO effectively reduced myostatin, Atrogin-1, and MuRF-1 transcripts and proteins, mitigating the atrophic process driven by CKD. This downregulation of key E3 ubiquitin ligases indicates the suppression of the ubiquitin–proteasome system, a primary driver of muscle protein degradation in CKD-associated sarcopenia. The findings corroborate previous studies identifying Atrogin-1 and MuRF-1 as crucial muscle atrophy mediators and highlight the pivotal role of myostatin as an upstream regulator [62,63,64]. Apart from the previously noted findings, there was a marked reduction in Collagen-I levels in skeletal muscle after MSTN-ASO treatment, suggesting an alleviation of muscle fibrosis—a secondary but critical aspect of CKD-induced sarcopenia. Improving both muscle mass and muscle quality further underscores the potential of MSTN-ASO as a comprehensive treatment approach.

A notable finding in this study was the muscle-specific effects of MSTN-ASO for the *MSTN* gene. Despite its pronounced efficacy in restoring muscle mass and function, MSTN-ASO did not improve renal function or morphology. The plasma levels of BUN and creatine remained elevated in treated CKD mice, and histological assessments showed persistent renal fibrosis and damage. This result suggests that MSTN-ASO selectively targets the *MSTN* gene in skeletal muscle without exerting direct reno-protective effects. This lack of renal recovery observed with MSTN-ASO treatment may be attributed to the predominant action of MSTN on muscle growth regulation and protein metabolism rather than involvement in renal tissue repair [65]. While skeletal muscle is the primary site of myostatin expression, and its presence has also been extended to various non-muscle tissues, such as adipose tissue and certain kidney structures, its expression level in the kidney is relatively low compared to that in skeletal muscle [66,67,68,69]. In addition, previous studies have indicated that MSTN inhibition enhances muscle regeneration by suppressing fibrosis and promoting satellite cell proliferation in skeletal muscle [29] (p. 2). In contrast, unlike some previous studies reporting systemic benefits of MSTN inhibition, including improvements in metabolic health and inflammation [59,60,61,70] (p. 10), our results indicate that the beneficial effects of MSTN-ASO are predominantly confined to skeletal muscle without a significant impact on kidney function or fibrosis. While no adverse effects of MSTN-ASO were observed in the present study, further investigation is necessary to elucidate the precise effects of MSTN-ASO on muscle as well as non-muscle tissue.

CKD-associated sarcopenia, also referred to as uremic sarcopenia, shows a progressive loss of muscle function and is driven by a combination of uremic toxins (indoxyl sulfate), and hallmark cytokines associated with the inflammatory process, including IL-1, IL-6, and TNF-α, along with oxidative stress, leading to elevated myostatin levels that exacerbate muscle wasting [14,71] (p. 2). Our findings confirm this pathological link, as CKD mice in the study displayed significantly upregulated myostatin levels alongside Atrogin-1 and MuRF-1, collectively driving muscle catabolism. The suppression of these mediators by MSTN-ASO demonstrates the key role of myostatin inhibition in halting the muscle-wasting cascade. Previous studies have collectively identified elevated myostatin expression as a key factor in uremic sarcopenia and outlined its inhibition as a therapeutic target [69,72,73,74] (p. 11). Our findings also support this association by providing in vivo data and bringing attention to myostatin inhibition by MSTN-ASO as a targeted gene therapy, reducing myostatin levels, resulting in muscle mass, and improving functional outcomes in CKD.

The MSTN-ASO used in our study, KMM001, is an ASO with a chimeric structure incorporating both 2′-OMe RNA and ENA with the phosphorothioate backbone. The chemistry of the 2′-OMe RNA and ENA with the phosphorothioate backbone is usually responsible for the robust exon-skipping activity, high stability, and high affinity for complementary RNA [75]. At present, Renadirsen, a chimeric ASO composed of 2′-OMe RNA and ENA, is being tested through clinical trial phases to determine its viability as a DMD intervention. In a single-time dose study where the preliminary dose range of Renadirsen was set as 1–30 mg/kg for skeletal and cardiac muscle tissues of mice, Renadirsen exhibited a strong exon-skipping effect compared with ENA-free ASOs at 10 mg/kg and 30 mg/kg, maintaining consistent exon skipping 1 week after a single dose [76].

Given these findings, KMM001 is highly anticipated to suppress MSTN in skeletal muscles and promote hypertrophy in atrophic muscle tissues. Consequently, it is screened as the most suitable antisense oligonucleotide to inhibit the myostatin gene. A key strength of our study is the comprehensive analysis of both functional and molecular outcomes following MSTN-ASO (KMM001) treatment. To our knowledge, this is the first in vivo study demonstrating the efficacy of KMM001 in CKD-induced muscle atrophy, extending previous in vitro findings [37] (p. 2). The dual evaluation of muscle mass restoration and molecular markers offers robust evidence supporting the therapeutic potential of MSTN-ASO in uremic sarcopenia. Compared with traditional strategies like exercise, nutritional supplements, and hormonal treatment, the results from this study suggest that MSTN-ASO is a potential gene-targeted therapy against CKD-induced muscle wasting. Unengaged with renal pathology, MSTN-ASO robustly promotes muscle hypertrophy and functional recovery by inhibiting MSTN and key mediators of muscle wasting. Such muscle-specific action represents a significant therapeutic advantage, enabling targeted muscle restoration in CKD.

However, the long-term safety and efficacy of MSTN-ASO must be further studied, especially if there are potential off-target effects, and their impact on other organ systems. Another area for future research involves exploring MSTN-ASO’s therapeutic potential in other sarcopenic conditions, including immobilization-induced, glucocorticoid-induced, and age-related sarcopenia.

In conclusion, our results firstly suggest that MSTN-ASO therapy can alleviate CKD-induced muscle atrophy by downregulating MSTN, Atrogin-1, and MuRF-1, key muscle degradation mediators, and by increasing the muscle CSA, mass, strength, and function. Our findings thus provide a strong foundation for future clinical translation by identifying potential novel gene-targeted therapies for CKD-associated sarcopenia, offering new hope for effective treatment strategies.

## 4. Materials and Methods

### 4.1. Animal Experiment

Our animal studies adhered rigorously to the ethical regulations defined by the 1964 Declaration of Helsinki and its subsequent modifications with the approval of the Shimane University ethics board (IZ5-48). Male C57BL/6J mice, aged 8 weeks, were sourced from CLEA, Inc. (Tokyo, Japan) and randomly sorted into the non-CKD (n = 8) and CKD (n = 24) groups. Non-CKD mice (considered as healthy controls) were fed a regular standard chow throughout the experimental period. On the other hand, the CKD group was fed a 0.2% adenine-supplemented diet for 4 weeks for CKD development. Then, CKD mice were divided into 3 subgroups: CKD (regular diet + 0.9% physiological saline, n = 8), CKD + Low-Dose ASO (regular diet + 25 mg/kg ASO, n = 8), and CKD + High-Dose ASO (regular diet + 50 mg/kg ASO, n = 8) and administered MSTN-ASO (KMM001) dissolved in 0.9% saline or a vehicle (0.9% physiological saline only) subcutaneously once a week (injection dose of 10 mL/kg) for 8 weeks. CKD (regular diet + 0.9% physiological saline) was used as a disease control. To identify the optimal treatment period for a weekly dosing regimen, after 3 weeks (at the beginning of 4 weeks) and 8 weeks of treatment, the treadmill test and forelimb grip test evaluated muscle function. Because 3 weeks of weekly dosing was not sufficient to improve muscle function (see Appendix A), the treatment period was extended to 8 weeks. Then, after 1 week of the last dose of MSTN-ASO, all mice were sacrificed, and we collected samples including blood, kidney, liver, heart, gastrocnemius, tibialis anterior, and soleus muscles, which were then measured for weight. Mice were housed with appropriate conditions, such as at 20–22 °C and 50% relative humidity, maintained in a regulated environment with a 12 h alternating light–dark regimen. An uninterrupted provision of standard chow was ensured for the mice, along with water. Each mouse’s body weight was monitored and recorded weekly. There was a pilot study where we first examined the in vivo effect of KMM001 on mice. However, this study was not performed to show the dose-dependent effect of KMM001. The dose of KMM-001 in this study was used to produce the sufficient effect expected, according to another in vivo study using another chimeric ASO (DS-5141b, Renadirsen, synthesized by KNC Laboratories Co., Ltd., Kobe, Japan) where subcutaneous injection was remarkably effective. Both KMM001 and Renadirsen are designed as 2’OMeRNA/ENA chimera with the phosphorothioate backbone [37] (p. 2). Their robust exon-skipping activity usually depends on the chemistry of this 2’OMeRNA/ENA chimera. It is notable that the preliminary dose range of Renadirsen was 1–30 mg/kg, and its single-time dose study showed robust exon skipping in healthy muscle at doses of 10 mg/kg and 30 mg/kg, where robust exon skipping persisted for 1 week following a single administration [76] (p. 11). Thus, it is assumed that subcutaneous administration of KMM001 once a week at a dose of 25 mg/kg will be as effective, and as expected, that was the case for KMM001 in the present experiment. However, we added a higher (50 mg/kg) dose of KMM001 just in case.

### 4.2. Treadmill Exhaustion Test

A treadmill exhaustion test was performed after 3 weeks and 8 weeks of treatment using a 2-lane treadmill (TMS, Melquest Ltd., Toyama, Japan). For the acclimation, the treadmill angle was set to 10° with a stimulatory shock of 0.3 mA, and the belt speed was adjusted to 50 rpm. Then, each mouse was allowed to run for 5 min, and acclimation was conducted for three consecutive days. In the final test, the mouse was allowed to run at 50 rpm with a 10° incline for 10 min. After that, the belt speed gradually increased by 10 rpm every 3 min until the mice could not run or until they reached the maximum speed of 200 rpm. The mouse was considered fully exhausted when remaining in the shock grid area for more than 20 s instead of having a stimulatory shock at the rear end. Each mouse’s total time, maximum speed for exhaustion, and total traveled distance were calculated upon reaching exhaustion.

### 4.3. Forelimb Grip Strength Test

After 3 weeks and 8 weeks of treatment, forelimb maximum force in each mouse was recorded with the help of a digital grip strength meter (GPM-101V; Melquest, Toyama, Japan). A metal bar was vertically attached to a digital force transducer. The mice grasped the bar with their forelimbs and were pulled back slowly, and the peak force released by the forelimbs was recorded using a force gauge as Kg. After resetting the force gauge to 0, the test was repeated twice, totaling three trials, and the average value was calculated.

### 4.4. Immunohistochemistry Staining

After dissection, the gastrocnemius muscle was directly placed in a cryovial, minimally immersed in O.C.T compound (Tissue-Tek, Sakura, Japan, Ctg#4583), and quickly frozen using isopentane after being rapidly chilled by liquid nitrogen and later securely kept in a −80 °C freezer. After that, the frozen tissue was cut into 7 μm thickness using a cryotome (Leica, Wentzler, Germany). The gastrocnemius muscle sections were collected on slides, and blocking was implemented by immersing the samples in a mixture of 5% goat serum together with 0.1% Triton-X 100 at a controlled ambient temperature for 1 h. After that, the overnight exposure of muscle tissue sections to primary antibody (Anti Laminin, Ctg#L9393, Sigma-Aldrich, Darmstadt, Germany, 1:25) was conducted at 4 °C, followed by 1–2 h incubation with goat-source anti-rabbit secondary antibodies such as Alexa Fluor 488 (Abcam, Cambridge, UK, Ctg#150077, 1:500). Stained muscle sections were photographed by a fluorescence microscope (Nikon Eclipse 80i, Tokyo, Japan). About 200 myofibers within a muscle section were outlined to evaluate the cross-sectional area and fiber size distribution by using ImageJ software (version 1.54).

### 4.5. Quantitative Real-Time PCR

The gastrocnemius muscles of mice were processed for total RNA extraction by employing RNA extraction reagent (Trizol, Invitrogen, Thermo Fisher Scientific, Waltham, MA, USA) in accordance with the manufacturer’s instructions. A Thermo Scientific NanoDrop OneC Cuvette Trace UV/Vis Spectrophotometer (Thermo Fisher Scientific, Waltham, MA, USA) was used to quantify the RNA concentrations of the samples. A total of 2 µg RNA served as the template for cDNA synthesis, facilitated by the RiverTraAce reverse transcriptase enzyme (Toyobo, Osaka, Japan). qPCR was performed, and expression profiles of distinct genes were elucidated via Thermal Cycler Dice by Takara, operating with software version 5.11 for real-time PCR. Normalization of gene expression was performed by calculating delta Ct values with GAPDH as the reference gene. The primer sequences were purchased from Hokkaido System Science Co., Ltd., after being determined from the prior literature, as documented in Arounleut et al. [77].

### 4.6. Western Blotting

Gastrocnemius muscle tissues were sonicated, and total protein was extracted with a pre-cooled RIPA buffer solution, as referenced in past publications [78]. We used 20×wt/vol of RIPA buffer to homogenize muscle tissues. Then, they were centrifuged, and supernatants were collected. The 3X sample buffer was mixed with the supernatant, and a protein sample ranging from 20 to 40 µg was separated via electrophoresis using 10% SDS polyacrylamide gel. Upon completion of separation, the protein was blotted onto a PVDF membrane (Millipore, Billerica, MA, USA), and a 1 h membrane incubation in 5% skim milk in TBST at RT was performed for blocking. Following blocking, the membrane was mixed with primary antibodies derived from a rabbit source against myostatin (Abcam, CT#ab203076), Atrogin-1 (Abcam, CT#168372), MuRF-1 (Proteintech, Chicago, IL, USA, CT#55456-1-AP), and Col-I (Abcam, CT#ab21286). After that, the membrane was immunoblotted with anti-rabbit Horseradish Peroxidase (HRP)-conjugated secondary antibody (1:4000, Ctg#HAF008, R&D, Minneapolis, MN 55413, USA). GAPDH (Cell signaling technology, Danvers, MA, USA, Ct#2118) was used as an internal control. For image acquisition and visualization, the Image Quant 800 detection system from Amersham (GE Healthcare, Little Chalfont, Buckinghamshire, UK) was employed. The expression level of each protein was quantified using ImageJ software and normalized to GAPDH expression.

### 4.7. Measurement of BUN and Creatinine Concentrations

After collecting 150 µL of blood from each mouse, the samples underwent centrifugation at a speed of 1500× *g* under 4 °C conditions for 10–15 min to obtain plasma. The BUN and creatinine concentration in plasma were measured by the biochemical auto-analyzer JCA-BM6070 (Nihon Deshi Co., Tokyo, Japan) according to the manufacturer’s protocol.

### 4.8. HE and Azan Staining

After trimming the excess fat, the dissected kidney was fixed in 4% paraformaldehyde at 4 °C for 12–16 h and then embedded in paraffin to make paraffin blocks. The paraffin blocks were then sectioned into 5 μm to make slides. The slides were deparaffinized using xylene (5 min × 3) and 100% or 99% ethanol (5 min × 3) and then stained with Meyer’s hematoxylin and eosin solution for histological study, and Azan stain was used to assess the collagenous fibers. Images were photographed using a fluorescence microscope (Nikon Eclipse 80i, Tokyo, Japan), and ImageJ software (version 1.54) was used to quantify the % of Azan-positive area.

### 4.9. Statistical Analysis

For statistical computations, GraphPad Prism 10.4.1 software was utilized as a statistical tool. While fiber size distribution is displayed as the mean, other results are depicted in the form of the mean ± SD. One-way ANOVA accompanied by Tukey’s multiple-comparison analysis was applied for group comparisons. Findings were interpreted based on *p*-values, with thresholds at 0.05, 0.01, and 0.001 for increasing levels of statistical significance.

## Figures and Tables

**Figure 1 ijms-26-03098-f001:**
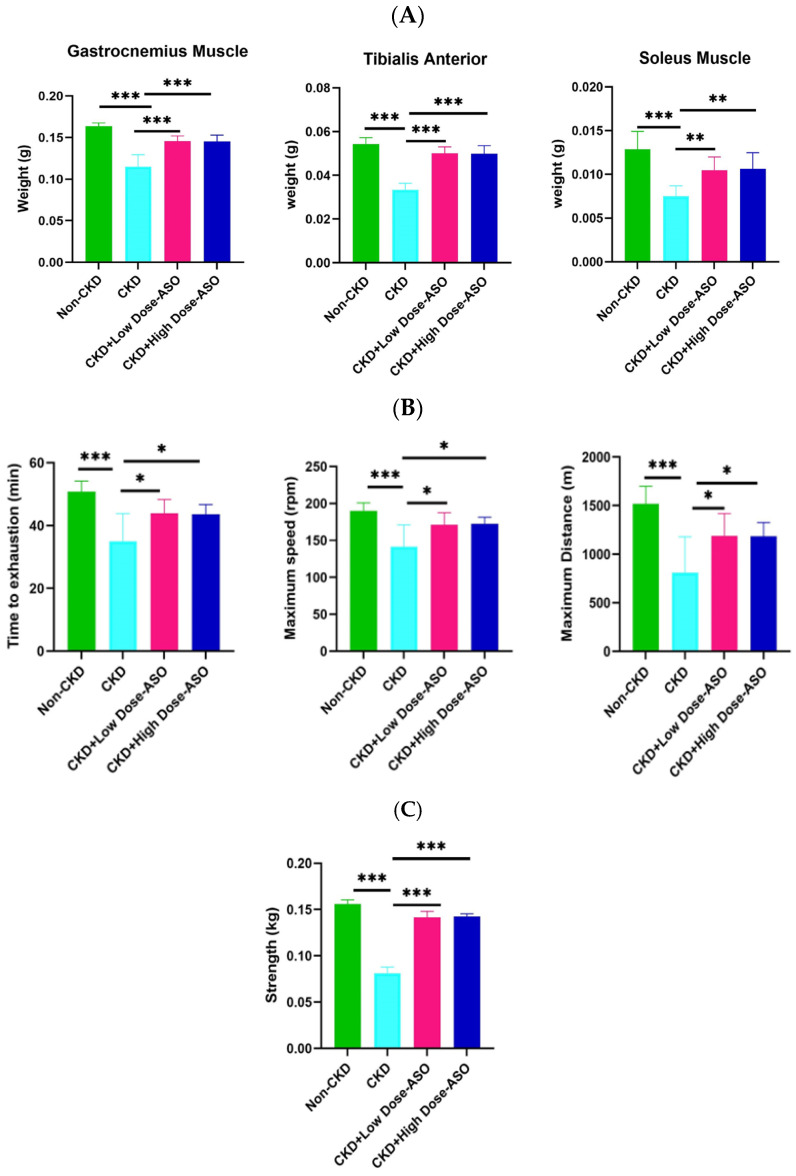
The effect of MSTN-ASO treatment on muscle mass, grip strength, and treadmill performance in CKD mice. (**A**) Weights of the gastrocnemius, tibialis anterior, and soleus muscle after 8 weeks of treatment with MSTN-ASO. (**B**) Mice’s time to exhaustion, maximum speed, and maximum distance were measured by the treadmill exhaustion test. (**C**) The peak grip strength of the forelimbs. The statistical approach involved one-way ANOVA combined with Tukey’s multiple-comparisons test. Results are depicted as the mean ± SD for 8 mice per group and interpreted based on *p*-values, with significance thresholds represented by * *p* < 0.05, ** *p* < 0.01, and *** *p* < 0.001.

**Figure 2 ijms-26-03098-f002:**
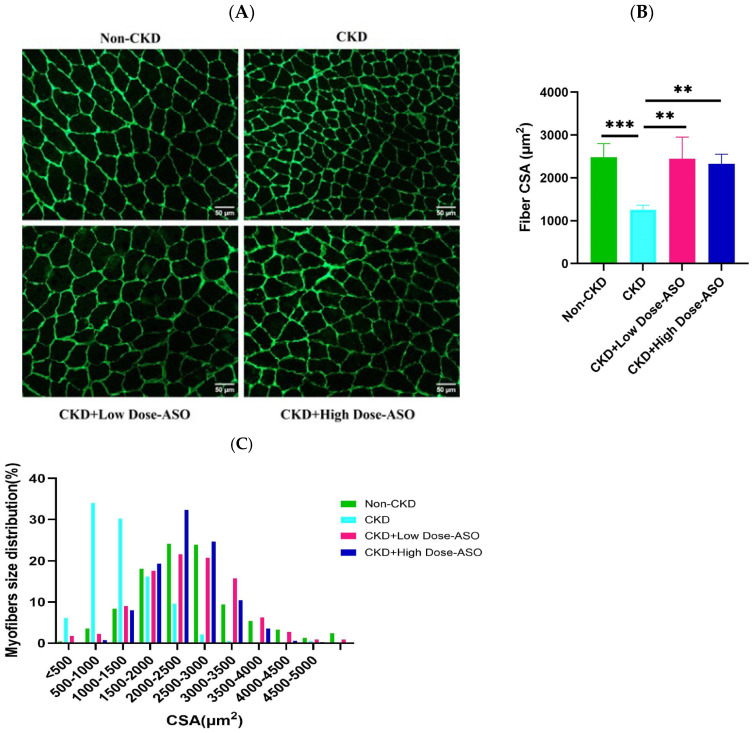
The effect of MSTN-ASO treatment on muscle atrophy. (**A**) Representative image of laminin staining of the mouse gastrocnemius muscle (scale bars: 50 µm, original magnification 20×). (**B**) Fiber cross-sectional area of the gastrocnemius muscle (mean ± SD, n = 4, nearly 200 individual myofibers were measured per mouse). (**C**) The distribution of myofiber size in the gastrocnemius muscle (mean, n = 4, nearly 200 individual myofibers were measured per mouse). The statistical approach involved one-way ANOVA combined with Tukey’s multiple-comparisons test. Results are depicted as the mean ± SD for 4 mice per group and interpreted based on *p*-values, with significance thresholds represented by ** *p* < 0.01 and *** *p* < 0.001.

**Figure 3 ijms-26-03098-f003:**
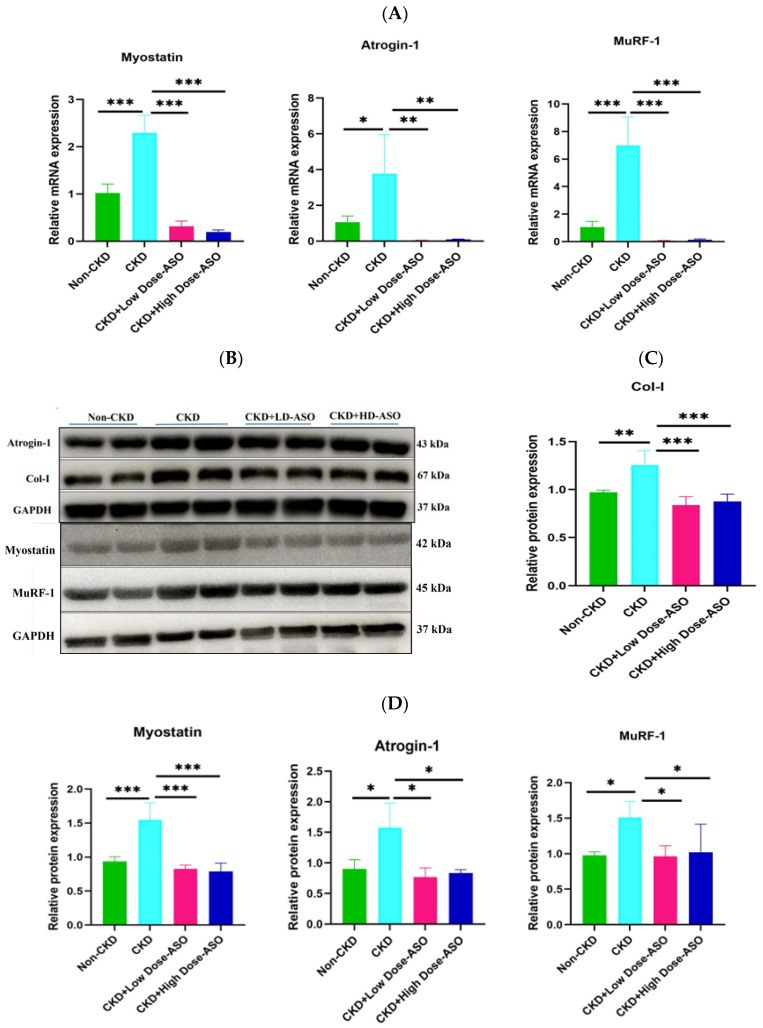
The effect of MSTN-ASO treatment on atrophy-related genes’ mRNA and protein expression. (**A**) The mRNA expression profiles of Myostatin, Atrophy F-box gene 1 (Atrogin-1) and Muscle RING-finger protein 1 (MuRF-1) within gastrocnemius tissue. (**B**) The Western blot protein band patterns for myostatin, Atrogin-1, MuRF-1, and Collagen-1. (**C**) The quantification of relative protein expression of Collagen-1. (**D**) The quantification of relative protein expression of Myostatin, Atrogin-1, and MuRF-1. The statistical approach involved one-way ANOVA combined with Tukey’s multiple-comparisons test. Results are depicted as the mean ± SD for 4 mice per group and interpreted based on p-values, with significance thresholds represented by * *p* < 0.05, ** *p* < 0.01, and *** *p* < 0.001.

**Figure 4 ijms-26-03098-f004:**
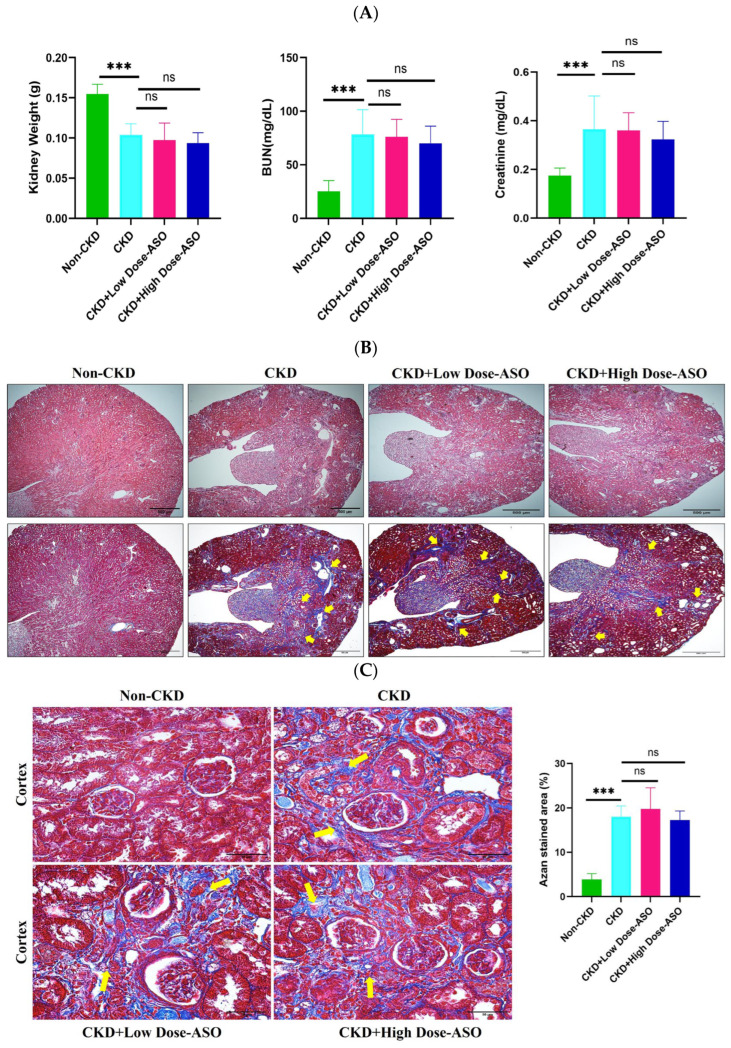
The effect of MSTN-ASO treatment on kidney function in CKD mice. (**A**) Kidney weight, BUN, and creatinine measured after MSTN-ASO treatment. (**B**) Representative images of HE and Azan staining of the whole kidney. Scale bars: 500 µm, original magnification 4×. (**C**) Representative images of Azan staining (cortex area, scale bars: 50 µm, original magnification 40×) (**left**) and quantitative analysis of the fibrotic area (**right**). Arrows indicate fibrotic areas. The statistical approach involved one-way ANOVA combined with Tukey’s multiple-comparisons test. Results are depicted as the mean ± SD for 4–8 mice per group, and interpreted based on *p*-values, with significance thresholds represented by *** *p* < 0.001 and ns: no significance.

## Data Availability

The original contributions presented in this study are included in the article/Appendix A; further inquiries can be directed to the corresponding authors.

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
