# Peer review of "Inhibiting Myostatin Expression by the Antisense Oligonucleotides Improves Muscle Wasting in a Chronic Kidney Disease Mouse Model"

_ijms, 2025, doi:10.3390/ijms26073098_

Round 1
Reviewer 1 Report
Comments and Suggestions for Authors
The submitted manuscript entitled Inhibiting myostatin expression by the antisense oligonucleotides improves muscle wasting in a chronic kidney disease mouse model is an excellent and interesting research topic. However, the authors should improve certain important points and present them properly :
1.- Abstract:
It is recommended that the authors start with a short introduction, taking care of the wording and not using too long sentences. The sentences should have a grammatical sequence and the appropriate ideas.
2.- Introduction:
It is recommended that the introduction should have the following sequence
Introduction, followed by the topic or point you want to address, then add 1 or 2 more recent works of other researchers related to your research work. Followed by your contribution and describe the importance of studying your work, e.g., we present. etc
3.- Materials and methods
The second section of any manuscript should be materials and methods.
In this section, authors are encouraged to describe the methodology used in a detailed and orderly fashion. This will ensure that the manuscript presents adequate and orderly reproducibility.
4.- Section 3, results and discussion
Here, the authors should adequately describe and detail their results and then discuss them. Comparing and contrasting them with related work by other authors is and will be of great importance, and finally, conclusively wording their results is crucial.
5.- Section 4 is the conclusion.
In this section, the authors should present, describe, and highlight the following points:
It is recommended that authors highlight their proposals, describe their advantages, and explain their most important results, applications, and future work.
6.- Authors are encouraged to make a graphical abstract ...
Other important aspects to be added to results and discussion:
Results:
a) The authors mention subcutaneous administration of MSTN-ASO, but they do not precisely describe the frequency of administration (daily, weekly?) and the total number of doses applied.
b) The authors do not mention whether a control group without CKD was used to compare the effects under normal conditions.
c) The authors also mention the measurement of muscle weight and the treadmill endurance test, which is appropriate. However, the study does not mention other functional or biochemical tests to support the authors' presentation.
d) Was MSTN expression measured in muscle tissue or plasma to confirm its effective inhibition?
e) Were changes in the activation of muscle signaling pathways such as Akt/mTOR or FoxO assessed?
f) Were adverse effects on other organs, such as the liver or kidneys, ruled out?
g) Did the treatment affect the expression of other genes related to muscle atrophy, such as MuRF1 or Atrogin-1?
h) Was it assessed whether the improvement in muscle function translates into a clinically relevant benefit for CKD patients?
i) Are treatment effects maintained after discontinuation of MSTN-ASO administration?
Reviewer 2 Report
Comments and Suggestions for Authors
Akhter et al. describes the use of MSTN-ASO (KMM-001) as a novel Myostatin inhibitor to ameliorate muscle wasting in an adenine induced CKD mice model. The authors describes the use of MSTN-ASO in a multidose treatment suppresses the expression of MSTN and the associated muscle specific E3-ubiquitin ligases Atrogin-1 and MuRF-1 – thereby leading to the increase in muscle mass, muscle mass CSA, and improvement in muscle strength and functions.
I believe this article greatly contributes to the field of oligonucleotide therapeutics, especially in the context of another way of inhibiting Myostatin to combat muscle wasting in CKD and other diseases. However, there are some revisions that the author needs to address to make this article suitable for publication. The comments are as follows:
- There is a discrepancy in the writing in the abstract vs the experimental section as to what constitutes a “Non-CKD” mice. The abstract mentions all procured mice were induced with CKD by feeding them a diet of 0.2% adenine for 4 weeks, after which “mice were randomly assigned to non-CKD (regular diet),…”. Over here, the non-CKD mice are basically CKD mice left to recover on a regular diet. On the contrary, the experimental section (4.1: Animal Experiment) mentions that C56BL/6J mice were “randomly sorted into the non-CKD and CKD groups. Non-CKD mice were fed a regular diet. On the other hand, in the CKD group, 0.2% adenine is mixed in the diet….”. In this scenario, the non-CKD mice are basically wild-type mice. Please clarify what was the exact manner non-CKD mice were determined. If the first scenario is correct, we need a wild-type mouse control in the experiments while if the second scenario is correct, we need a control for CKD mice on a regular diet (to see how much restoration of muscle function is observed after replacement of the CKD inducing diet with a regular diet).
- How long after the last dose were the mice sacrificed for analysis? Additionally, why was subcutaneous injection preferred over IV administration for this study? Was there a direct comparison done between the two methods to determine the best route of administration?
- How does the biodistribution of the chimeric ASO KMM-001 look like?
- How were the low (25 mpk) and high (50 mpk) doses determined? We barely see any dose response in any of the experiments included in the study, where it seems the 25 mpk low dose is enough to show maximal effect.
- There are no experiments demonstrating the results of a single dose study and/or a duration study (with a single dose) to elucidate the correct dosing regimen used in this report. The single dose study also would give an idea about the tolerability of the compound in mice.
Line 244-245 mentions “The dose of KMM-001 in this study was decided to get the sufficient effect expected, according to the previous studies [43].” However, the Reference 43 is about DMD where DMD specific ASOs were used and not KMM-001. It becomes more important that the results from a single dose study/ies are included in this study since this is the first mice study done with the compound after in vitro studies – according to the author.
- How much muscle wasting is seen in cardiac muscles in CKD mice model? Only effects on the skeletal muscles is described in this study.
- Why is that despite reduction in MSTN levels (thereby preventing muscle damage/degradation), we do not observe an increase in serum BUN and creatinine levels which are usually positively correlated with MSTN levels?
- Why does the kidney histopath for the “CKD+ High Dose-ASO” show a significantly higher level of fibrosis (as determined by Azan statin) in the uploaded “Original Blots/Gels” image despite the quantification in Figure 4C showing it to be similar?
- How does the approach of using an ASO compare to that of an siRNA since there have been a few reports utilizing siRNA for inhibiting MSTN?
The English is good and comprehensible.
Round 2
Reviewer 1 Report
Comments and Suggestions for Authors
abstract
the authors have improved their abstract
However, you should improve your fluency in writing and the precision of some terms, making it more attractive. They should also emphasize the novelty and clinical implications of their study.
- Section 2 contains materials and methods, and section 3 includes results and discussion. section 4 conclusions
In this section (3), the authors should explain the results obtained, contrast them with related research, and be conclusive in each.
The author section 2.1.
The authors stated that, as an initial result, they continue to present poor scientific writing. The authors mention that the results are consistent with previous studies, but they do not discuss them in detail.
Also, no exact values for increases in muscle mass or physical performance are presented (only p-values).
It is not reported whether there were any side effects in the treated mice.
They must be strengthened with a better contrast to previous literature and more detailed statistical data.
The authors' graphs should be labeled orderly; they do not show them properly (Figure 3).
Although the graphics and images are different, they should be presented and labeled appropriately, with good image quality and even the same size.
authors' sections 2.4
Although studies on CKD models are cited, the effects of MSTN-ASO are not compared with other treatments for renal dysfunction.
No exact values for kidney weight, BUN, or creatinine are presented; only significant differences are mentioned.
It is not explained why MSTN-ASO does not improve renal function.
Better contrast with the literature and a more in-depth interpretation of negative findings is lacking.
Reviewer 2 Report
Comments and Suggestions for Authors
I thank the authors for acknowledging the comments and their appropriate responses. While my previous comments are satisfactorily answered, there remains some minor yet critical revisions that needs to be made within the manuscript. After these changes are made, this manuscript should be significantly improved, be engaging to the readers and be suitable for publication. Please see the comments below:
- Please add a couple of sentences within the manuscript and in the experimental section how you arrived at 25 mpk (especially for a multidose study) as the low dose and 8 weeks of weekly dosing as the optimal treatment. You refer to the previous DMD paper being the basis for choosing this dose, but over there you only used 10 mpk of Renadirsen for 4 weeks for the multidose study. It would also be good to include a tissue concentration data (in various muscle groups) after 8 weeks of treatment if you already have it.
- Please add a statement within the manuscript stating that the treatment with 0.2% adenine is enough to irreversibly induce CKD, and that after the 4 weeks of 0.2% adenine supplemented diet, switching to regular diet do not reverse the CKD conditions. And please add references too for that statement.
- Line 45: Add references after this statement. Also add a couple sentences briefly describing the current therapies being investigated for treatment of CKD.
- Please use arrows to indicate some of the aberrant pathological features on the representative kidney histopathology image for CKD mice (Figure 4b/ Figure 4c).
- Line 236, Line 277-278: The statements made about “tissue specificity” of the compound KMM-001 are misleading. Those statements should be removed/rephrased to make it clear that it is not tissue specific.
Although there is no biodistribution data for the current compound under investigation, similar ASO-Renadirsen (as referenced by the author themselves and also mentioned as an answer to my previous question in the response letter) shows major accumulation in clearance tissues like kidney and liver.
Since MSTN is not expressed in kidney, you are seeing muscle-specific effects for this gene. This does not make the compound muscle tissue-specific.
- Line 281-283: Another statement on MSTN-ASO having a “favorable safety profile and no-off target adverse events” are misleading too, and these sentences should be rephrased.
This is because we are looking at a CKD mice model, where there is already extensive kidney damage. It is difficult to gauge further damage incurred by the compound (KMM-001). Also, since these ASO primarily gets accumulated in kidney, there is a good possibility of having basophilic granules due to compound accumulation in lysosomes which are not good in the long term.
To make any conclusive statement for KMM-001 to “have a favorable safety profile”, the authors would need to investigate at the toxicity profile in WT mice (non-CKD) (both single and multidose studies) and analyze all urinary (and/or serum) biomarkers that are associated with kidney damage and analyze the kidney histopathology data.
Since that is not the case here in this study, those sentences needs to be modified to remove such indications (favorable safety profile and no off-target events).
Round 3
Reviewer 1 Report
Comments and Suggestions for Authors
The manuscript presented by the authors is an interesting research topic; however, in order to be published, the authors must take care of minor but important details.
As mentioned in previous reviews, section 2 corresponds to materials and methods, section 3 corresponds to results and discussion. At this point, the authors, in addition to writing logically, scientifically, and straightforwardly, should their results with related research (previous) results and finally be conclusive. Finally, section 4 is the conclusion; here, the authors must describe their research results correctly, highlighting these results, advantages, disadvantages, and proposals, and be conclusive.
However, authors are recommended to cite in their manuscript correctly, because after evaluating the similarity report, authors present sentences and/or expressions that are not cited, which may be plagiarism. Finally, authors are advised not to use very old references, older than 6 years.
Therefore, the manuscript is similar to 40%, and it is recommended that the authors check and structure their manuscript adequately based on the comments.
Authors are advised user reference not major that 6 years.
This is because of several references that are older than 10 years.
